# Learning Implicit Credit Assignment for Cooperative Multi-Agent Reinforcement Learning

**Meng Zhou**[*]     **Ziyu Liu**[*]     **Pengwei Sui**     **Yixuan Li**     **Yuk Ying Chung**
The University of Sydney

## Abstract

We present a multi-agent actor-critic method that aims to implicitly address the credit assignment problem under fully cooperative settings. Our key motivation is that credit assignment among agents may not require an explicit formulation as long as (1) the policy gradients derived from a centralized critic carry sufficient information for the decentralized agents to maximize their joint action value through optimal cooperation and (2) a sustained level of exploration is enforced throughout training. Under the centralized training with decentralized execution (CTDE) paradigm, we achieve the former by formulating the centralized critic as a hypernetwork such that a latent state representation is integrated into the policy gradients through its multiplicative association with the stochastic policies; to achieve the latter, we derive a simple technique called *adaptive entropy regularization* where magnitudes of the entropy gradients are dynamically rescaled based on the current policy stochasticity to encourage consistent levels of exploration. Our algorithm, referred to as LICA, is evaluated on several benchmarks including the multi-agent particle environments and a set of challenging StarCraft II micromanagement tasks, and we show that LICA significantly outperforms previous methods.

## 1   Introduction

Many complex real-world problems such as autonomous vehicle coordination [3], network routing [55], and robot swarm control [17] can naturally be formulated as multi-agent cooperative games, where reinforcement learning (RL) presents a powerful and general framework for training robust agents. Though single-agent RL algorithms can be trivially applied to these environments by treating the multi-agent system as a single actor with a joint action space, the limited scalability and the inherent constraints on agent observability and communication in many multi-agent environments necessitate *decentralized* policies that act only on their local observations. A straightforward approach to learn decentralized policies is to train the agents independently, but the simultaneous exploration of the agents often results in non-stationary environments where learning becomes highly unstable. To this end, previous work relies on a standard paradigm known as *centralized training with decentralized execution* (CTDE) [34, 25, 37, 10, 28, 4], where the independent agents can access additional state information that is unavailable during policy inference.

However, one major challenge of CTDE in cooperative settings is *credit assignment*, which refers to the task of attributing a global, shared reward from the environment to the agents' individual actions. Solving the credit assignment problem explicitly may give useful insights into which agents or agent actions were responsible for the collective reward signal and may thus substantially facilitate policy optimization, but doing so is often nontrivial since the interactions between the agents and the environment can be highly complex. A notable approach is to assess individual agent actions by calculating *difference rewards* against a certain reward baseline [48, 36, 10], but these methods can

---

[*]Equal contribution. Author ordering is random. Correspondence to Meng Zhou <mzho7212@gmail.com> and Ziyu Liu <kenziyuliu@outlook.com>.

be inefficient as they require separate estimations for the baselines and can become less effective with complex cooperation behaviors. Another line of approach is to represent the global state-action value as a (rule-based or learned) aggregation of the individual state-action values [46, 37, 45, 28]. A representative method is QMIX [37], which achieves *implicit* credit assignment by learning a non-linear *mixing network* that conditions on the global state and maps the individual agent $Q$-values into the joint action $Q$-value estimate. While these methods allow more complex credit assignment, the capacity of the value mixing network is still limited by the monotonic relationships between the joint and the individual $Q$-values. Their extensions to continuous action spaces may also require additional strategies that can compromise performance and/or complexity.

In this work, we propose a policy-based algorithm called LICA for learning implicit credit assignment that aims to address the above limitations. LICA is closely related to the family of *value gradient* methods [7, 52, 15, 43, 14, 26, 2] where policies are directly optimized in the direction of the approximate state/action value gradients. Apart from applying the framework to multi-agent settings, our key contribution is to extend the concept of value mixing [46, 37, 28, 45] to *policy mixing*, where the centralized critic is formulated as a hypernetwork [11] that maps the current state information into a set of weights which, in turn, mixes the individual action vectors into the joint action value estimate. Compared to previous policy-based methods such as [10, 27, 18], this practical formulation introduces an extra latent state representation into the policy gradients to provide sufficient information for learning optimal cooperative behaviors without explicit credit assignment strategies. It also trivially achieves higher expressiveness than value mixing methods as there are no inherent constraints on the mixing weights of the centralized critic. Following the above motivation, we also explore an alternative training regime where continuous approximations of action samples from the stochastic policies are replaced with explicit action distribution parameters to provide more information about the agents' behaviors during policy optimization. While the resulting action value gradients can be less accurate, we observe that policies can learn stably and often converge faster to better solutions.

One notable challenge for policy-based algorithms is maintaining consistent levels of exploration to prevent premature convergence to sub-optimal policies [53, 30, 1]. Many existing methods [42, 13, 18, 27, 49] address this with entropy regularization, where the policy entropy is added to the training objective to favor more stochastic actions. While widely adopted, we argue that the vanilla form of entropy regularization could be ineffective for this purpose due to the undesirable curvature of the entropy function derivative. To this end, we further propose a simple technique called *adaptive entropy regularization* where the magnitudes of entropy gradients are inversely adjusted based on the policy entropy itself. We show that this allows easier tuning of the regularization strength and more consistent levels of policy stochasticity throughout training.

We benchmark our methods on two sets of cooperative environments, the Multi-Agent Particle Environments [27] and the StarCraft Multi-Agent Challenge [39], and we observe considerable performance improvements over previous state-of-the-art algorithms. We also conduct further component studies to demonstrate that (1) compared to difference reward based credit assignment approaches (e.g. [10]), LICA has higher representational capacity and can readily handle environments where multiple global optima exist, and (2) our adaptive entropy regularization is crucial for encouraging sustained exploration and can lead to faster policy convergence in complex scenarios.

## 2 Related Work

**Explicit Credit Assignment.** In general, explicit methods provide strategies for attributing agent contributions that are at least provably locally optimal [24]. COMA [10] is a representative method that uses a centralized critic to estimate the counterfactual advantage of an agent action, but it quickly becomes ineffective with complex cooperation behaviors. SQDDPG [49] provides a theoretical framework where credit assignment is based on an agent's approximate *marginal contribution* as it is sequentially added to the agent group. While theoretically justified, this formulation assumes an initial oracle scheduling the agents with global observation that is often unavailable in practice.

**Implicit Credit Assignment.** In contrast, implicit methods do not purposely assess the individual actions against a certain baseline, and most previous methods address credit assignment by directly learning a value decomposition from the shared reward signal into the individual component value functions [46, 37, 45, 28]. An earlier work is VDN [46], where the value decomposition is linear and the state information is ignored during training. QMIX [37] presents an improvement by conditioning

a hypernetwork on the global state for a non-linear mixing of the individual action-values, but it is still limited by the monotonicity constraint of its mixing weights. QTRAN [45] attempts to address these limitations with a provably more general value factorization, but it imposes computationally intractable constraints that, if relaxed, can lead to poor empirical performance [28]. In contrast, the policy-based LICA is practical and has high capacity as no additive or monotonic constraints exist.

**Model-Free Value Gradients and Deterministic Policy Gradients.** As our method learns the decentralized policies by backpropagating the joint action value, it is related to the family of value gradient [52, 7, 8, 15, 14] and deterministic policy gradient (DPG) [43, 26, 27, 49] algorithms. On a high level, both classes of methods aim to improve the policy by directly ascending the gradients of (approximate) state-values ($V$) or action-values ($Q$) via chain rule [14]. In particular, DPG methods assume deterministic policies to efficiently estimate policy gradients, and they can be considered the deterministic limit of the model-free variant of stochastic value gradient (i.e. SVG(0)) [15], a type of value gradient methods that deploys *reparameterization* (e.g. [23, 38, 20]) to similarly allow backpropagation of action value gradients. Among existing methods, MADDPG [27] extends the DPG framework to multi-agent settings, where each agent learns a deterministic policy from joint action value gradients and maintains a separate (one for every agent) centralized MLP critic network for learning different reward structures. In contrast, our method learns stochastic policies from the centralized critic (one for all agents) formulated as a hypernetwork, which is crucial for better utilization of the state for learning complex agent cooperation without credit assignment strategies.

**Entropy Regularization** is a common technique for improving policy optimization by inducing more stochastic actions and possibly a smoother objective landscape [1], where many existing methods [53, 12, 30, 40, 42, 13, 18, 27, 49] augment the training objective with a weighted entropy loss term. Others also consider, e.g., decaying schedules for the regularization strength [32, 1]. As mentioned in Section 1, we argue that its common form could be ineffective at keeping consistent exploration levels that are often required for uncovering optimal joint actions in challenging environments.

# 3 Methods

## 3.1 Preliminaries and Notations

In this work we primarily focus on learning policies in discrete action spaces, though extensions to continuous action spaces are possible. We consider a fully cooperative multi-agent task with $n$ agents $\mathcal{A} = \{1, ..., n\}$ as a Dec-POMDP [33] defined by a tuple $G = (S, U, P, r, Z, O, n, \gamma)$. At each time step $t$, each agent $a$ chooses an action $u_t^a$ from its action space $U_a \in \{U_1, ..., U_n\} \equiv U$, forming a joint action $u_t \in (U_1 \times \cdots \times U_n) \equiv U^n$. $P(s_{t+1}|s_t, u_t): S \times U^n \times S \to [0, 1]$ is the state transition function where $S$ is the set of true states and $s_t \in S$ is the state at time $t$. $r(s_t, u_t): S \times U^n \to \mathbb{R}$ is the reward function yielding a shared reward $r_t$ for $u_t$, and $\gamma \in [0, 1)$ is the discount factor. We consider partially observable settings, so each agent $a$ acts on its local observation $z_t^a \in Z$ drawn from the observation function $O(s_t, a): S \times \mathcal{A} \to Z$ with observation space $Z$. To select an action, each agent $a$ follow its stochastic policy $\pi^a(u_t^a|z_t^a): Z \times U_a \to [0, 1]$, and the agents' joint policy $\pi$ induces a joint action value function $Q^\pi(s_t, u_t) = \mathbb{E}_\pi[R_t|s_t, u_t]$ where $R_t = \sum_{t'=t}^{T} \gamma^{t'-t} r_{t'}$ is the discounted accumulated reward with finite horizon $T$. Our goal is to find the optimal joint policy $\pi^*$ such that $Q^{\pi^*}(s_t, u_t) \geq Q^\pi(s_t, u_t)$ for all $\pi$ and $(s_t, u_t) \in S \times U^n$. We also use $Q^\pi(s_t, u_t)$ and $Q^\pi(s_t, u_t^1, ..., u_t^n)$ interchangeably and use $\tau = (s_1, z_1, u_1, r_1, s_2, ...)$ to denote a trajectory. Function approximations of $\pi$ and $Q^\pi$ are parameterized by $\theta = \{\theta_1, ..., \theta_n\}$ and $\phi$ respectively.

## 3.2 LICA: Learning Implicit Credit Assignment for Cooperative Environments

In this section, we present a new method called LICA that aims to implicitly address multi-agent credit assignment under shared rewards in fully cooperative environments. Our key motivation is that if the decentralized policies can be directly optimized to maximize the true joint action value function, then any converged policies should have acquired (at least locally) optimal cooperative behaviors without *explicitly* formulating a credit assignment among agents, which can be unrealistic and unscalable in complex scenarios. In practice, if a trained critic $Q_\phi^\pi(s, u)$ provides a close approximation of $Q^\pi$ for the current $\pi_\theta$, then finding an end-to-end differentiable optimization setting where the policies simultaneously improve along the joint action value gradients $\nabla_\theta Q_\phi^\pi(s, u)$ should act as a proxy for finding optimal credit assignment strategies (e.g., studied in COMA [10]).

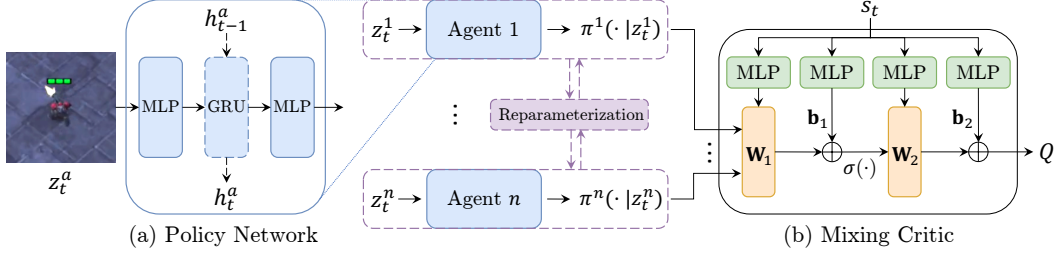

$h_{t-1}^a$

$z_t^a$

$h_t^a$

(a) Policy Network

$z_t^1 \rightarrow$ Agent 1 $\rightarrow \pi^1(\cdot | z_t^1)$

Reparameterization

$z_t^n \rightarrow$ Agent $n$ $\rightarrow \pi^n(\cdot | z_t^n)$

$s_t$

(b) Mixing Critic

Figure 1: **Overview of LICA**. (a) Architecture for the decentralized policy networks. (b) Architecture for the centralized mixing critic network. Dashed components are situational. Best viewed in color.

To this end, a simple approach taken by MADDPG [27] is to formulate $Q_\phi^\pi$ as an MLP that maps the concatenation of the state and actions (or more generally, their learned representations) into the joint $Q$ estimate. In LICA, we instead propose to formulate $Q_\phi^\pi$ as a *hypernetwork* [11] that maps the state into a set of weights which, in turn, maps the concatenated action vectors into the $Q$ estimate. We refer the resulting critic as the *mixing critic*, which is illustrated in Fig. 1. The key insight of this structural design is that vector concatenation in an MLP critic implies that the representations of the state $s$ and the joint action $u$ are associated through *addition*; with a critic hypernetwork, the additive association is converted to a *multiplicative* association through which an extra representation of the current state is integrated into the action value gradients to provide more information about the environment for the agents to address credit assignment implicitly through learning.

Concretely, let us first consider the partial derivative $\frac{\partial Q_\phi^\pi}{\partial \theta} = \frac{\partial Q_\phi^\pi}{\partial u} \frac{\partial u}{\partial \theta}$ for updating the policy parameters $\theta$. When comparing the structures of the mixing critic (abbreviated as $C_{\text{Mix}}$) and the MLP critic ($C_{\text{MLP}}$), we only need to focus on $\partial Q_\phi^\pi / \partial u$ as $\partial u / \partial \theta$ is unrelated to the critic. Let us then consider the general case where both $C_{\text{MLP}}$ and $C_{\text{Mix}}$ operate on learned representations of $s$ and $u$, denoted as $f_s(s)$ and $f_u(u)$ with non-linear $f_s$ and $f_u$ (e.g. MLP at input heads). In both cases, we can write:

$$\frac{\partial Q_\phi^\pi}{\partial u} = \frac{\partial Q_\phi^\pi}{\partial \nu} \frac{\partial \nu}{\partial u}, \text{ with } \nu = \begin{cases} f_s(s) + f_u(u) \\ f_s(s) f_u(u) \end{cases} \text{ and } \frac{\partial \nu}{\partial u} = \frac{\partial \nu}{\partial f_u} \frac{\partial f_u}{\partial u} = \begin{cases} \frac{\partial f_u}{\partial u}, & \text{for } C_{\text{MLP}}. \\ f_s(s) \frac{\partial f_u}{\partial u}, & \text{for } C_{\text{Mix}}. \end{cases} \quad (1)$$

Here, $\nu$ is the first *mixed* representation of $f_s(s)$ and $f_u(u)$ in the critic network before subsequent non-linearity: for $C_{\text{MLP}}$, it is the hidden representation immediately after the concatenation and the subsequent linear transform; for $C_{\text{Mix}}$, it is the representation immediately after the linear transform by the state-dependent hyper-weights $\mathbf{W}_1$ in Fig. 1 (b), which can be considered $f_s(s)$. As the rest of the critic network $g(\nu) = Q$ is learned, non-linear and non-interpretable in both cases, the key difference from Eq. 1 is thus that $C_{\text{Mix}}$ changes the composition of $\partial Q_\phi^\pi / \partial \theta$ and introduces an extra, direct state representation $f_s(s)$ to facilitate policy learning towards implicit credit assignment.

In fact, similar techniques for altering gradient properties with multiplicative association were explored in the computer vision [9] and sequence modeling [54] literature, where element-wise multiplications between hidden features can act as a "gating mechanism" for information flow. In our case, while we do not focus on gating and the change in the gradient does not by itself guarantee a better credit assignment, we empirically show that the better utilization of the state made available from CTDE by $C_{\text{Mix}}$ provides a better basis for learning credit assignment and better joint policies.

**Critic Learning.** We train $Q_\phi^\pi$ on-policy with a practical variant of generalized advantage estimation [41] and TD($\lambda$) [47] adapted from [10, 39], with targets $y_t^{(\lambda)}$ defined recursively up to $T$, as:

$$\min_\phi \mathbb{E}_{\tau \sim \pi_\theta(\tau)} \left[ \left( y_t^{(\lambda)} - Q_\phi^\pi(s_t, u_t) \right)^2 \right], \quad y_t^{(\lambda)} = r_t + \gamma \left( \lambda y_{t+1}^{(\lambda)} + (1-\lambda) Q_\phi^\pi(s_{t+1}, u_{t+1}) \right). \quad (2)$$

One could replace $y_t^{(\lambda)}$ with samples of $R_t$ from reward trajectories as unbiased estimates of $Q^\pi$, though such estimates may have higher variance that scales with $T$. One may also use a *target* critic network $Q_{\phi^-}^\pi$ with periodic updates $\phi^- \leftarrow \phi$ to improve overall learning stability [31, 10, 39].

**Policy Learning.** With $Q_\phi^\pi \approx Q^\pi$, we train decentralized policies simultaneously to maximize the following objective, where $\mathcal{H}$ is the entropy regularization term for each agent (more in Section 3.3):

$$J(\theta) = \mathbb{E}_{\tau \sim \pi_\theta(\tau)} \left[ Q_\phi^\pi(s_t, u_t) + \mathbb{E}_a \left[ \mathcal{H} \left( \pi_{\theta_a}^a(\cdot | z_t^a) \right) \right] \right]. \quad (3)$$

In particular, we learn stochastic policies over deterministic policies because: (1) they can better handle state aliasing [14] and are harder to exploit [6] under partial observability; (2) there exists

POMDPs where stochastic policies can be arbitrarily better than any deterministic policies [44, 19, 21]; and (3) they admit on-policy learning and exploration [15, 14]. With small and discrete action spaces, we can compute the policy gradients from Eq. 3 directly with backpropagation as the expected $Q$ over the actions can be obtained as $\pi_\theta(\cdot|z)^\top Q^\pi_\phi(s, \cdot)$ which is differentiable w.r.t. to $\theta$. However, in the multi-agent case where the joint action space $U^n$ grows exponentially, this is impractical and the expectation over the actions is estimated by sampling from the stochastic policies, which is not directly differentiable. Drawing connections to SVG(0) [15], one can backpropagate through samples by *reparameterizing* the stochastic policies as deterministic functions of independent noises; in the discrete case, this can be achieved using Gumbel-Softmax [20], a continuous, differentiable relaxation of the Gumbel-Max reparameterization for the categorical action samples. Here, we also consider an unconventional alternative previously explored in [52, 51] where we directly feed the *action distribution parameters* of each agent (e.g. mean and variance of a Gaussian policy and action probabilities of a discrete policy), denoted as $\pi^a_t := \pi^a_{\theta_a}(\cdot|z^a_t)$, as inputs to the critic for computing approximate policy gradients. Specifically, the policy of an agent $a$ is updated using the following:

$$\frac{\partial J(\theta_a)}{\partial \theta_a} \approx \mathbb{E}_{\tau \sim \pi_\theta(\tau)} \left[ \frac{\partial Q^\pi_\phi(s_t, \pi^1_t, ..., \pi^n_t)}{\partial \pi^a_t} \frac{\partial \pi^a_t}{\partial \theta_a} + \frac{\partial \mathcal{H}(\pi^a_t)}{\partial \theta_a} \right]. \tag{4}$$

This formulation has several notable properties: during policy training, it can be considered a special case of DPG as the agents *deterministically* map their observations into the action parameters [14], allowing end-to-end differentiability without continuous approximations of action samples; the exploration of the action parameters is likewise influenced by entropy regularization, as is the previous case with sampling from the stochastic policies; and in contrast to the previous case, the action distributions contain more information (than sampled actions) about the agents' behaviors [52], which intuitively can improve the learning efficiency towards better joint actions. In the discrete case, the approximation in Eq. 4 induces a slight inconsistency between how the critic is used (probabilistic actions as inputs) and trained (special cases with probabilities 0 or 1); in practice, however, we show that policies can indeed converge (and often faster) to good solutions in complex scenarios.

### 3.3 Adaptive Entropy Regularization

As LICA is policy-based, it shares a common limitation that insufficient exploration may lead to premature convergence to poor policies. This is particularly notable when agents are acting in real-world environments where sampling a large number of diverse trajectories can be expensive or impractical. To this end, previous methods rely on an additional entropy term $\mathcal{H}(\pi^a(\cdot|z^a)) = \beta H(\pi^a(\cdot|z^a)) = \beta \mathbb{E}_{u^a \sim \pi^a}[-\log \pi^a(u^a|z^a)]$ for agent $a$ weighted by some constant $\beta$ during optimization. The rationale is that penalizing a low policy entropy resulting from under/over-confident actions should encourage sustained exploration throughout training. However, in practice we observe that this may not be the case: $\beta$ often has high sensitivity in complex environments and a tiny nudge could lead to drastically different entropy trajectories; moreover, once policies start to converge (i.e. drop in entropy), the same regularization term may not encourage further exploration.

We hypothesize that these issues stem from the *curvature* of the entropy function derivative which in turn influences the magnitudes of its policy gradients on different action probabilities. Concretely, let us first consider the $k$-class action probability vector $p^a = \pi^a(\cdot|z^a)$ of an agent $a$ with the number of feasible actions $k > 2$. The derivative of $\mathcal{H}$ with respect to $p^a$ is given by:

$$d\mathcal{H} = \left[ \frac{\partial \mathcal{H}}{\partial p^a_1}, \cdots, \frac{\partial \mathcal{H}}{\partial p^a_k} \right] = [-\beta(\log p^a_1 + 1), \cdots, -\beta(\log p^a_k + 1)] \tag{5}$$

assuming natural logarithm. We denote $d\mathcal{H}_i := -\beta(\log p^a_i + 1)$ and note that $d\mathcal{H}_i$ depends only on the $i$-th action probability $p^a_i$ when $k > 2$. In particular, since $d\mathcal{H}_i$ is log-shaped, the gradient magnitudes for dampening high-confidence actions (large $p^a_i$) are disproportionately small compared to that for encouraging low-confidence actions (small $p^a_i$). While favoring the latter should help achieve the former, we argue that in practice, insufficient penalties for over-confidence can still result in agents sticking to a subset of high-confidence actions while only updating the action probabilities of other actions during stochastic optimization. On this basis, a reasonable approach is to manually induce a larger penalty (gradient magnitudes) for large $p^a_i$. To this end, we propose a practical technique called *adaptive entropy regularization* where we dynamically control the magnitudes of the entropy gradients such that they are inversely proportional to the policy entropy itself during training:

$$d\mathcal{H}_i := -\xi \cdot \frac{\log p^a_i + 1}{H(p^a)} \tag{6}$$

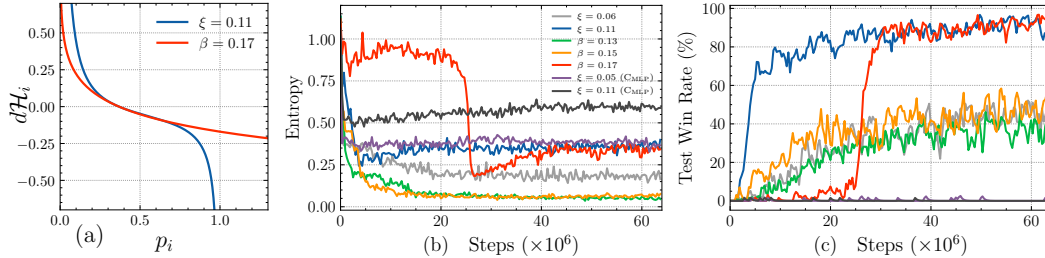

Figure 2: **Effects of *adaptive entropy regularization* and *mixing critic***. (a) Plots of the entropy function derivative before (with $\beta$) and after (with $\xi$) applying Eq. 6. (b-c) Entropy trajectories and test win rates on StarCraft II `5m_vs_6m`. Runs not labeled with $C_{MLP}$ use a mixing critic ($C_{Mix}$).

with constant $\xi$ controlling regularization strength. We visualize its values and curvature in Fig. 2 (a)[2], and note that large action probabilities now incur greater penalties from the entropy regularization.

This formulation has a clean interpretation that the regularization strength previously controlled by the constant $\beta$ is now *adaptive* based on the policy stochasticity: if the policy exhibits deterministic behavior (i.e. low entropy), then the strength is scaled up by dividing a smaller term (the entropy); and if the policy is too stochastic, the strength is scaled down by dividing a larger term. In the multi-agent case, the entropy measure is defined per-agent, though in practice one could simplify and average it across all agents. While we do not provide a convergence analysis here, we show (Section 4.3.2, Fig. 2) that in practice, policies can indeed reach a stochasticity equilibrium (with entropy level determined by $\xi$), which can possibly help maintain a smooth objective landscape [1] and aid policy convergence. Possible future work may also generalize Eq. 6 by replacing $H(p^a)$ with other policy stochasticity measures, such as a multiplicative inverse of the $L^2$ norm of the difference between $p_a$ and a uniform action probability vector, that may lead to different empirical or theoretical outcomes.

# 4 Experiments

## 4.1 Multi-Agent Particle Environments

We first evaluate our algorithm against previous state-of-the-art methods on two common multi-agent particle environments [27]: Predator-Prey and Cooperative Navigation.

**Predator-Prey.** In this environment, 3 cooperating agents control 3 predators to chase a faster prey by controlling their velocities with actions [up,down,left,right,stop] within an area containing 2 large obstacles at random locations. To focus on cooperative behavior, we follow [49] where the prey acts randomly. Each predator only observes its own velocity and position and its displacement from the prey, obstacles, and other predators. The goal is to capture the prey with a minimal number of steps, and the shared reward is the minimum distance between the prey and any of the predators. The game terminates when the prey is captured and an additional positive shared reward is given.

**Cooperative Navigation.** In this environment, $n$ agents and $n$ landmarks are initialized with random locations within an area, and the agents must cooperate to cover all landmarks by controlling their velocities with actions [up,down,left,right,stop]. Each agent only observes its own velocity and displacement from other agents and the landmarks, and the shared reward is the negative sum of displacements between each landmark and its nearest agent. Agents must also avoid collisions, as each agent incurs $-1$ shared reward for every collision against other agents. The environment uses $n = 3$ by default, and we also extend the difficulty with $n = 5$.

**Training Settings.** For all environments, agents are trained for 5000 episodes, each has a maximum of 200 steps and may end early for Predator-Prey. We follow the open-source implementations from [49] for the baseline algorithms, including COMA [10], independent DDPG [26] (IDDPG), MADDPG [27], and SQDDPG [49]. Each algorithm is trained with the same 5 random seeds and the mean and standard deviation of the goal metric (steps per episode, and mean reward over all timesteps and agents, respectively) throughout training are reported. See Supplementary for further settings.

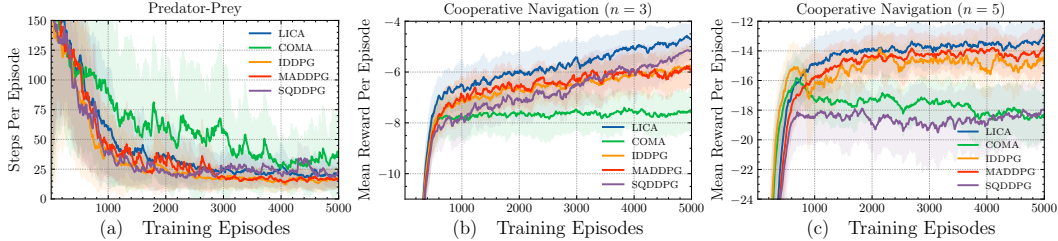

Figure 3: Comparing performance during training in **Multi-Agent Particle Environments**. (a) Predator-Prey. (b-c) Cooperative Navigation with $n = 3$ and $n = 5$ respectively.

**Results.** Fig. 3 reports the results for the Particle Environments. We first observe that LICA performs on par with or better than previous state-of-the-art methods, verifying its effectiveness. We can also see that LICA consistently outperforms COMA and MADDPG, confirming its capacity for credit assignment. Moreover, $n = 5$ for Cooperative Navigation increases the task difficulty as SQDDPG now underperforms and MADDPG yields better relative performance possibly due to the growth of its complexity with $n$. For Predator-Prey, LICA achieves similar convergence speed and performance compared to other methods, suggesting a saturation with the environment likely due to the random (instead of learned) prey. Despite achieving competitive results, we believe that the particle environments may not be sufficiently challenging to validate our methods, given that IDDPG, which completely ignores cooperation and/or credit assignment, is already competitive.

## 4.2 StarCraft II Micromanagement

**Configurations.** StarCraft II (SC2) provides a rich set of heterogeneous units each with diverse actions, allowing extremely complex cooperative behaviors among agents. We thus further evaluate LICA on several SC2 micromanagement tasks from the SMAC [39] benchmark, where a group of mixed-typed units controlled by decentralized agents needs to cooperate to defeat another group of mixed-typed enemy units controlled by built-in heuristic rules with "difficult" setting; the battles can be both symmetric (same units in both groups) or asymmetric. Same as previous work [10, 37, 46, 4], we used the default environment settings from SMAC. Each agent observes its own status and, within its field of view, it also observes other units' statistics such as health, location, and unit type; agents can only attack enemies within their shooting range. A shared reward is received on battle victory as well as damaging or killing enemy units. Each battle has step limits set by SMAC and may end early. We consider 6 battle maps grouped [39] into **Easy** (2s3z, 1c3s5z), **Hard** (5m_vs_6m, 2c_vs_64zg), and **Super Hard** (MMM2, 3s5z_vs_3s6z) against 5 baseline methods using their open-source implementations based on PyMARL [39]: COMA [10], LIIR [4], VDN [46], QMIX [37], and QTRAN [45].

**Training Settings.** The inherent differences across the baseline methods and their training procedure (e.g. on/off-policy learning for policy/value-based methods) make it difficult to juxtapose them without introducing extra components (e.g. importance sampling [29, 50, 5] for off-policy policy evaluation) that could alter the baseline performance. To this end, [39] scales down both the batch size and the number of batch updates for on-policy methods in an effort to align their sample efficiency against off-policy methods with experience replay; while reasonable, this makes it hard to attribute any poor performance of on-policy methods to either the poor training conditions (particularly high variance from small batch sizes and insufficient gradient steps) or the underlying algorithmic limitations. To focus on the latter, we instead follow [4] where all methods use batches of 32 episodes generated with parallel runners to align the number of both batch updates and environment steps, which is in total 32 million steps for Easy (due to fast convergence) and 64 million for Hard and Super Hard scenarios. All methods also use GRU modules (Fig. 1) and share the parameters of the individual agent networks. Performance is evaluated every 320k env steps with 32 test episodes, and we report the median, 1st, and 3rd quartile win rates across five random seeds for every method in every map. See Supplementary for further environment and training details and battle visualizations.

**Results.** Fig. 4 reports the results across all 6 scenarios.[3] We first observe that LICA achieves competitive win rates across homogeneous and heterogeneous groups (e.g. 5m_vs_6m and 2s3z),

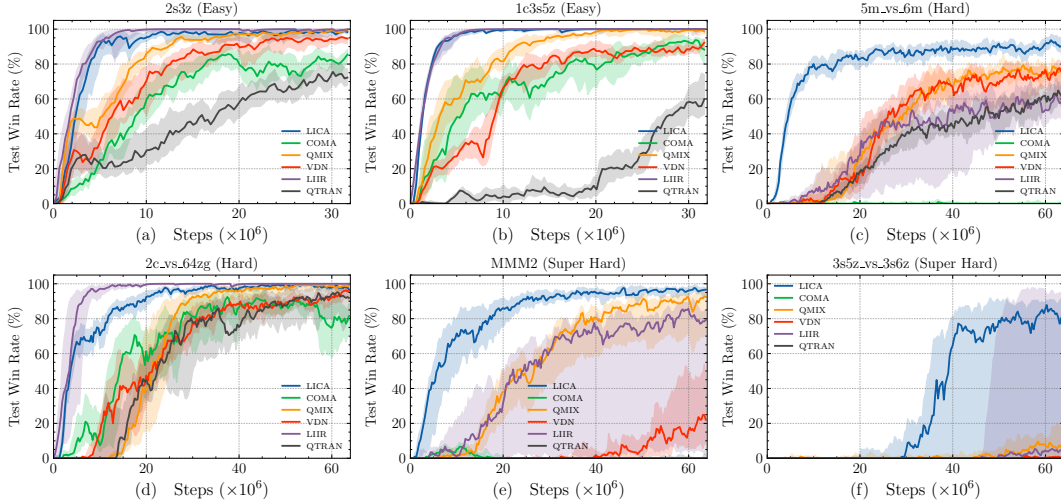

Figure 4: Performance comparison across various scenarios for **StarCraft II micromanagement**.

symmetric and asymmetric battles (e.g. `1c3s3z` and `3s5z_vs_3s6z`), and varying number of units (e.g. 2 units in `2c_vs_64zg` and 10 units in `MMM2`), underpinning its robustness. LICA also tends to give more consistent results than other methods across different difficulties. All methods solve Easy scenarios reasonably well, though the increased joint action diversity of `1c3s5z` over `2s5z` slows down convergence for certain methods while LICA is unaffected due to its high capacity. For the asymmetric `5m_vs_6m` (Hard), basic agent coordination alone such as "focus firing" [10, 37, 39, 4] no longer suffices and consistent success requires extended exploration to uncover complex cooperative strategies such as pulling back units with low health during combat. In `2c_vs_64zg`, while LICA outperforms most methods and reaches nearly perfect win rate, it underperforms LIIR in convergence speed; this is possibly because an optimal control of only 2 Colossus units may prioritize individual performance over cooperation where LIIR could benefit from its explicit formulation of an individual reward. Nevertheless, on the most challenging scenarios `MMM2` (involving 10 units of 3 types) and `3s5z_vs_3s6z` (where the extra enemy Zealot gives significant advantage), LICA's considerable performance improvements confirm the effectiveness of our proposed methods.

## 4.3 Component Studies

### 4.3.1 1-Step Traffic Junction

To verify the credit assignment capacity of LICA, we introduce a simple 1-step environment where multiple optimal joint actions exist. The motivation here is to use a minimal scenario to illustrate the limited capacity of explicit credit assignment methods based on difference rewards for learning opti-

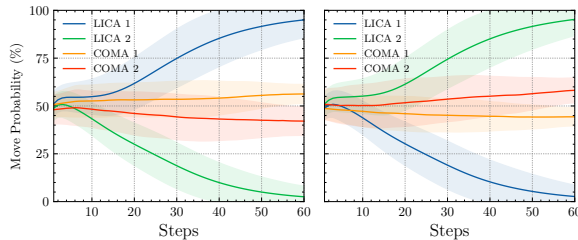

Figure 5: 1-Step Traffic Junction agent `move` probabilities throughout training for the two optimal outcomes.

mal policies, and having multiple optima is one such scenario that is also common in the real world (e.g. traffic management). The 1-step game involves two vehicles controlled by two separate agents attempting to pass a traffic junction in order. Each agent can either `pass` or `wait`; if both try to `pass` or `wait` at the same time, they receive a shared reward of 0; otherwise they receive a shared reward of 1, resulting in two optimal joint actions. It is worth noting that the simplicity of this 1-step game obviates most key aspects that differentiate on/off-policy learning (e.g. replay buffers and separate target/behavior nets) and focuses only on the mechanism for credit assignment. Taking COMA [10] as a representative example, difference reward based agents will expect a counterfactual advantage of 0 over the four possible joint actions; that is, if the centralized critic captures the true joint action value function, then on average, COMA agents perform as good as random agents. In practice, however, imperfect value functions and/or stochastic gradient descent (as opposed to full-batch gradient descent to consider all possible counterfactual actions) do allow agents to converge to optimal policies. To experiment, we train LICA and COMA agents for 20k random initializations each with 60 steps and

show the mean and standard deviation of the agent action probabilities for the two optimal outcomes in Fig. 5. The fast policy convergence and low policy entropy of LICA verifies the effectiveness of its implicit credit assignment.

### 4.3.2 Ablation Experiments

We further perform ablation experiments on SC2 `5m_vs_6m` (Hard) to validate each proposed component in LICA.

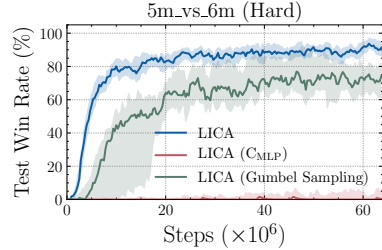

Figure 6: Ablations for different LICA components on SC2 `5m_vs_6m`.

**Mixing Critic.** We first consider replacing the centralized mixing critic ($C_{Mix}$) with an MLP critic ($C_{MLP}$, Section 3.2) which maps the concatenated representations of the state and actions into the joint $Q$ estimate. In Fig. 2 (b-c), we perform two runs (labeled with $C_{MLP}$) against the best LICA run ($\xi = 0.11$) with different entropy regularization strengths, where $\xi = 0.11$ ($C_{MLP}$) provides a controlled comparison and $\xi = 0.05$ ($C_{MLP}$) leads to a similar entropy level. We also provide repeated runs with $\xi = 0.05$ ($C_{MLP}$) in Fig. 6. The poor performance with $C_{MLP}$ in all experiments (see also Supplementary) clearly indicates the necessity of the mixing critic.

**Policy Optimization with Action Distribution Parameters.** We next consider how policy learning using action distribution parameters (i.e. probabilistic actions, Eq. 4) instead of sampled actions may affect performance. In Fig. 6, we report the results of LICA agents trained with "Straight-Through" Gumbel-Softmax [20] on `5m_vs_6m` (results on more SC2 maps are deferred to Supplementary). Intriguingly, despite the critic may not provide good estimates for probabilistic actions, we observe that LICA policies trained with Eq. 4 often converge faster and more stably to better optima than those learned with Gumbel sampling (though we believe the latter can improve its stability with well-crafted temperature schedules). One useful intuition is that action distribution parameters are more informative about the agents' behaviors than sampled actions [52], and as "softer" (yet deterministic to the agent observations) inputs to the critic, they could lead to less variance in the resulting value gradients [16, 20]; from a learning perspective, both of these properties can be favorable to policy optimization. Despite the empirical evidence supporting this argument, however, it would be interesting to develop further theoretical insights into this training regime in future work.

**Adaptive Entropy Regularization.** We further perform an ablation study to validate the proposed adaptive entropy regularization scheme and report the results in Fig. 2. We first observe that the adaptive control of regularization strength (with $\xi$) leads to sustained levels of policy stochasticity throughout training, which is not the case with vanilla entropy regularization (with $\beta$). This property also makes tuning regularization strengths easier: increasing the strength for LICA from the sub-optimal $\xi = 0.06$ where the policy converges prematurely to the optimal $\xi = 0.11$ leads to a *predictable* change in the entropy trajectory, which is also the case for LICA ($C_{MLP}$) when $\xi$ is raised from 0.05 to 0.11; in contrast, when tuning the default entropy regularization ($\beta \in \{0.13, 0.15, 0.17\}$), slight changes in initial strength can lead to either similar or vastly different entropy trajectories. We also see that while $\beta = 0.17$ can reach similar win rates as $\xi = 0.11$, the agent exploration levels are drastically different before and after converging to exploitable policies (i.e. leap in win rates); in contrast, adaptive regularization encourages a consistent stochasticity level where the policies improve steadily and converge faster. We believe that given a large $\beta$ and sufficient training, the vanilla formulation can indeed lead to optimal policies due to LICA's inherent high capacity, though the empirical results verify the benefits of the proposed adaptive entropy regularization.

## 5 Conclusions and Future Work

This paper presents LICA, a new actor-critic method that aims to implicitly address the credit assignment problem for fully cooperative multi-agent RL problems. LICA learns the decentralized policies by directly ascending the approximate joint action value gradients, and it makes use of two key components: a centralized critic hypernetwork that integrates a state representation into the policy gradients, and an adaptive entropy regularization scheme to dynamically control the policy stochasticity for encouraging extended and consistent agent exploration throughout training. Compared to previous methods, LICA has a simpler and a more general formulation, has higher capacity for credit assignment, and performs competitively in practice. For future work, we aim to improve its sample-efficiency by integrating off-policy training techniques; extend it for multi-agent continuous control; and explore further theoretical properties of adaptive entropy regularization.

## Broader Impact

As many complex real-world problems can be formulated as cooperative multi-agent games, this work provides an effective approach to these problems. For example, decentralized agents can be applied to network routing optimization to speed up transmission, traffic management with autonomous vehicles to maximize traffic flow, and efficient package delivery with swarms of drones to reduce delivery costs. However, since our method relies on deep neural networks to implicitly attribute shared outcomes of the agent group to the individual agents, it faces the "black box problem" where behaviors of the individual agents may not be rational or interpretable from the human perspective. Furthermore, when maximizing a shared reward in multi-agent cooperative settings without considering the status of the individual agents, ethical issues may arise when the optimal joint actions require sacrificing certain agents. Using the task of traffic management with autonomous vehicles as an example, maximizing the total traffic volume could lead to indefinite delays for a subset of the vehicles.

## Funding Disclosure

The authors declare that they have no conflict of interest.

## Acknowledgments

We thank the anonymous reviewers and Tabish Rashid for useful discussions and valuable feedback for improving the earlier version of this work.

## Footnotes

[2]Note that $k = 2$ is a special case where the action probabilities are interdependent and Eq. 5 does not apply. For visualization purposes, Fig. 2 (a) follows Eq. 5, but note that our analysis applies to $k > 2$.

[3]Code is available at https://github.com/mzho7212/LICA. At the time of writing, our experiments are based on the latest PyMARL framework which uses SC2.4.10 while the original results reported in [39] uses SC2.4.6. As indicated by the original authors, performance is not always comparable across SC2 versions.

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
