[Supplementary Material]

# A1  Implementation Details

Our code for the StarCraft II micromanagement tasks (Fig. 4 of the main paper) is available at https://github.com/mzho7212/LICA.

## A1.1  StarCraft Multi-Agent Challenge

For our experiments on StarCraft II micromanagement, we follow the setup of the StarCraft Multi-Agent Challenge (SMAC [39]).[4] We used the open-source implementations of our baseline algorithms, including COMA [10], LIIR [4], VDN [46], QMIX [37], and QTRAN [45] based on the PyMARL framework.[5] For scenarios shown in Fig. 4 of the main paper:

- 2s3z is a symmetric battle where two heterogeneous groups of 2 Stalkers and 3 Zealots battle against each other; Stalkers are ranged units and Zealots are melee units (short attack range).
- 1c3s5z is a symmetric battle where two heterogeneous groups of 1 Colossus, 3 Stalkers, and 5 Zealots battle against each other; Colossus units are ranged, more endurable than Stalkers and Zealots, and their attacks cover a small area instead of a single unit.
- 5m_vs_6m is an asymmetric battle where 5 Marines battle against 6 Marines.
- 2c_vs_64zg is an asymmetric battle where 2 Colossi battle against 64 Zerglings, which are melee units with lower health and lower attack damage.
- MMM2 is an asymmetric battle where 1 Medivac, 2 Marauders, and 7 Marines battle against 1 Medivac, 3 Marauders, and 8 Marines. Medivacs are healer units with no outgoing damage, and Marauders are ranged units with higher health and attack damage compared to Marines but with slower attack speed (longer cooldown between attacks).
- 3s5z_vs_3s6z is an asymmetric battle where 3 Stalkers and 5 Zealots battle against 3 Stalkers and 6 Zealots. The extra enemy Zealot makes this scenario far more challenging than the symmetric version 3s5z.

The discrete action space of each agent consists of move[direction], attack[agent_id], stop, and noop. direction is one of east, west, north, or south, and only dead agents can take the action noop. For special healer units (e.g. Medivac in MMM2), the action heal[agent_id] replaces attack[agent_id]. All agents can only use the attack[agent_id] action within their shooting range, which can be different (and often smaller) compared to their field of view, thus disabling macro-actions such as *attack-move*. All automatic actions, such as automatically attacking enemies in range, are disabled such that agents need to learn their strategies without any guidance. The feature vector of each observable agent within an agent's field of view includes distance, relative_x, relative_y, health, shield, and unit_type. Agents can observe terrain features such as terrain height and walkability. The agent_id of each agent is also included in the local agent observations.

The global state information, on top of the agent local observations and only available to the centralized critic, includes the positions of all agents relative to the centre of the map, the energy of Medivacs (for healing other units), cooldown of the rest of the allied units (minimum time interval between consecutive attacks), and the last actions performed by all the agents. All observation vectors and state vectors are normalized by their max values. At each time step, agents receive a shared joint action reward based on the damage dealt or the enemies killed, and there is an additional reward for winning the battle. See SMAC [39] for further environment details.

Most of our training hyperparameters follow PyMARL [39]. Critical hyperparameters, such as the entropy regularization coefficient ($\beta$ for vanilla entropy regularization and $\xi$ for adaptive entropy regularization), are tuned either manually or with grid search. The network structures of the policy networks and the mixing critic are implemented as illustrated in Fig. 1 of the main paper: the agent networks have two FC layers and a GRU layer in between, and the mixing critic produces two weight matrices and biases for mapping the concatenated agent action probabilities into the joint action value ($Q$) estimate. All hidden layers have 64 units. The batch size $b$ is set to 32. Adam [22] optimizer is used, with initial learning rate 0.0025 for the policy networks and 0.0005 for the mixing critic. We

follow LIIR [4] where all methods align on the batch size $b$, the number of batch updates, and the total number of environment steps (summed across all $b$ parallel runners). The reward discount factor $\gamma$ is set to 0.99, and $\lambda$ for critic training is set to 0.8. We also clip gradients at $L^2$ norm of 10, and we use a target mixing critic, which is periodically updated every 200 gradient steps on the critic, to stabilize training. The regularization strength $\xi$ for adaptive entropy regularization is set to 0.11 for `5m_vs_6m`, 0.03 for `3s5z_vs_3s6z`, and 0.06 for all other scenarios. We use ReLU for all activation functions. See our released source code for additional training details.

### A1.2 Multi-Agent Particle Environments

For both particle environments (Predator-Prey and Cooperative Navigation), we adapt their open-source implementation from the original authors [27] and the training framework provided by [49], which can be accessed at the particle environments repo[6] and the SQDDPG repo[7] respectively. These repositories provide the pre-set environment hyperparameters such as the size of the agent/obstacle spawn regions, the positive shared reward for capturing the prey in Predator-Prey, and the collision penalty between agents in Cooperative Navigation. Note that the prey agent in Predator-Prey is replaced with a random agent with uniform action probability distribution.

For Fig. 3 of the main paper, we follow the default settings from SQDDPG [49]: for Predator-Prey and Cooperative Navigation respectively, the decentralized policy networks are implemented as 1-hidden layer MLPs with 128 and 32 hidden units; agents do not share policy network parameters; batch size $b$ is set to 128 and 32; discount factor $\gamma$ is set to 0.99 and 0.9 with critics trained using simple one-step TD error. We use target networks to stabilize training, which are updated every 200 training iterations. Adam is used as the optimizer with different initial learning rates for different methods, and we use a learning rate of 0.0003 for policy learning and 0.0003 for critic learning. The strength of adaptive entropy regularization $\xi$ is set to 0.1, 0.1, and 0.2 for Predator-Prey, Cooperative Navigation $n = 3$, and $n = 5$, respectively. During training, all methods also used the same set of 5 random seeds for the repeated runs.

### A1.3 Training Details/Pseudocode

We use PyTorch [35] all implementations. The pseudocode for LICA is summarized in Algorithm 1. In the earlier version of this work, we included an extra hyperparameter $k$ to represent the number of gradient steps to take for every batch of $b$ episodes when training the mixing critic (i.e. an inner loop for Eq. A7); in the updated version, we have removed it for all methods to avoid confusion (i.e. effectively $k = 1$), re-tuned all relevant hyperparameters (including the regularization coefficients $\beta$ and $\xi$), and re-run and reproduced all related experiments (e.g. Fig. 2).

To implement adaptive entropy regularization, the PyTorch-style pseudocode for obtaining the entropy loss term (for maximizing entropy) can be summarized as:

```
# treat policy entropy as a constant with ".item()"
adaptive_coeff = entropy_coeff / entropy.item()
entropy_loss = - adaptive_coeff * entropy
```

where `entropy` is the policy entropy and `entropy_coeff` is the constant $\xi$ controlling the regularization strength.

## A2 Additional Ablation Experiments

We extend the ablation experiments in Fig. 6 of the main paper to two other Hard/Super Hard StarCraft II micromanagement scenarios (`2c_vs_64zg` and `MMM2`), and we report the results in Fig. A1. We observed that LICA consistently outperforms the ablations, and, confirming our discussions in Section 4.3.2 of the main paper, LICA agents trained with Gumbel sampling give competitive performance but tend to converge slower with less stability while those trained with an MLP critic ($C_{\text{MLP}}$) tend to perform poorly. We also provide an architecture sketch for the MLP critic in Fig. A2.

**Algorithm 1** Optimization Procedure for LICA

---

1: Randomly initialize $\theta$ and $\phi$ for the policy networks and the mixing critic respectively.
2: Set $\phi^- \leftarrow \phi$.
3: **while** not terminated **do**
4:     Sample $b$ episodes $\tau_1, ..., \tau_b$ with $\tau_i = \{s_{0,i}, z_{0,i}, u_{0,i}, r_{0,i}, ..., s_{T,i}, z_{T,i}, u_{T,i}, r_{T,i}\}$.
5:     **for** episode $i = 1$ to $b$ **do**
6:         **for** timestep $t = T$ to $1$ **do**
7:             Compute the targets $y_{t,i}^{(\lambda)}$ according to Eq. 2 using target critic network $Q_{\phi^-}^{\pi}$.
8:         **end for**
9:     **end for**
10:    # Critic Update
      Update the mixing critic by descending the gradient according to Eq. 2:

$$\nabla_\phi \frac{1}{bT} \sum_{i=1}^{b} \sum_{t=1}^{T} \left( y_{t,i}^{(\lambda)} - Q_\phi^\pi \left( s_{t,i}, u_{t,i}^1, ..., u_{t,i}^n \right) \right)^2. \tag{A7}$$

11:    # Policy Update
      Update the decentralized policy networks by ascending the gradients according to Eq. 3 and Eq. 4, with entropy gradients adjusted according to Eq. 6 (see also Section A1.3) :

$$\nabla_\theta \frac{1}{bT} \sum_{i=1}^{b} \sum_{t=1}^{T} \left( Q_\phi^\pi \left( s_{t,i}, \pi_{\theta_1}^1(\cdot|z_{t,i}^1), ..., \pi_{\theta_n}^n(\cdot|z_{t,i}^n) \right) + \frac{1}{n} \sum_{a=1}^{n} \mathcal{H} \left( \pi_{\theta_a}^a(\cdot|z_{t,i}^a) \right) \right). \tag{A8}$$

12:    **if** at target update interval **then**
13:         Update the target mixing critic $\phi^- \leftarrow \phi$.
14:    **end if**
15: **end while**

---

Figure A1: Additional ablation experiments on two StarCraft II Hard/Super Hard scenarios.

## A3    Qualitative Analysis

We visualize and analyze two of the best battles performed by LICA agents on StarCraft II `5m_vs_6m` (**Hard**) and MMM2 (**Super Hard**) respectively. Demo videos are available at https://github.com/mzho7212/LICA.

In `5m_vs_6m` (5 Marines vs 6 Marines), we observe that LICA agents demonstrate several interesting micromanagement techniques that are often used by proficient human players. For example, LICA agents learned *stutter stepping*[8], which is a tactic where a unit moves right after its attack to best utilize its attack interval (time gap between its consecutive attacks) and maximize its mobility. Apart from *focus-firing* where LICA agents simultaneously focus on individual enemy units, they also learned to build strategic formations (e.g. Fig. A3) where high health Marines are moved to the front to attract the attacks of the heuristics-based enemy Marines that often prioritize closer targets. At the same time, low health Marines are moved frequently (and without sacrificing their damage through stutter stepping) to encourage enemy Marines to change their attack targets. Note that these

Figure A2: Architecture sketch for LICA with an MLP critic ($C_{MLP}$, Section 3.2).

Figure A3: Example agent formation by LICA (red units on the left) in StarCraft II `5m_vs_6m` where Marines with high health move forward while the dying Marine is pulled back. Green squares on the floor are agent `move` targets.

cooperative strategies require the agents to reason about higher level battle dynamics despite acting on their local observations in a decentralized manner.

On the other hand, `MMM2` (1 Medivac + 2 Marauders + 7 Marines vs 1 Medivac + 3 Marauders + 8 Marines) features units with different functionalities (Medivac is a healer unit), health (Marauder is more endurable), attack damages, and attack speeds (Marauders has a higher damage than Marines, but their attacks are slower). This allows more complex cooperative strategies between the agents. From the demo video, we first observe that LICA agents learned to prioritize killing enemy Marines because they are brittle but have higher DPS (damage per second) overall. In particular, the agents learned that the enemy Marines attacking the ally Medivac should be targeted first. We also observe that LICA agents learned to pull Marauders forward despite their longer shooting range compared to Marines; this is because Marauders are more durable and can afford to divert possible enemy attention and take more incoming damage. Moreover, unlike ally Marines, the two Marauders did not focus-fire the enemy Marines but instead chose their own targets with high health; we believe that this is to prevent *overkill* (dealing more damage than necessary to kill a unit) because their slow projectile and attack speeds imply that their attacks are harder to plan and easier to miss on enemy units that are already being focus-fired. Furthermore, we observe that the ally Medivac learned to switch between multiple ally units and prioritize units with low health. In particular, it learned to sacrifice (by not healing) a dying Marauder unit to focus on other units that could still be saved; this allows the battle to be won with only one Marauder killed while all other units remain.

## Footnotes

[4]https://github.com/oxwhirl/smac

[5]https://github.com/oxwhirl/pymarl

[6]https://github.com/openai/multiagent-particle-envs/blob/master/multiagent/scenarios (`simple_tag.py` and `simple_spread.py`).

[7]https://github.com/hsvgbkhgbv/SQDDPG/

[8]https://liquipedia.net/starcraft2/Stutter_Step