[Reviews · NeurIPS 2020]

Review 1

Summary and Contributions: This paper considers fully cooperative multi-agent tasks modeled as Dec-POMDPs and provides a new algorithm, Learning Implicit Credit Assignment (LICA), to train decentralized policies via centralized training with decentralized execution. LICA has two significant features: (a) a policy-mixing-network is proposed for credit assignment in a new implicit way; (b) an adaptive entropy loss is introduced into the policy gradient computation to further improve explorations during training. The evaluation of LICA is performed across several benchmark problems and compared with a set of state-of-the-art multi-agent RL approaches. The results show that LICA outperforms all the baselines and demonstrates the advantage of the adaptive entropy regularization. However, there are still significant issues with the method and several concerns about the results.

Strengths: The architecture of LICA is the first contribution of this work, where agents’ stochastic policies are concatenated first and then passed into a mixing network, whose weights are leaned from a ground truth state to output a joint Q-value as the centralized critic. This policy mixing network is certainly an interesting idea that extends the mixing network of QMIX and the architecture in MADDPG to stochastic policies. The adaptive entropy regularization is a novel idea. The strength of the entropy loss for encouraging exploration is now adaptive based on the stochasticity of each agent’s policy, such that when an agent’s policy is a near-deterministic one, the entropy loss will have a higher weight to further encourage the exploration. The advantage of having this adaptive feature is demonstrated in the results versus using a fixed weight on the entropy loss.

Weaknesses: The first essential issue in LICA algorithm is that the definition of the centralized value-function is not clear. In particular, what exactly is the proposed value function is trying to approximate? During training, this centralized value function is trained conditioned on a sampled joint action (Eq.3), while during policy updating, it is used in a way that conditions on the concatenation of the probability over actions output by each agent’s policy. Due to this inconsistency in the input of the value-function, this critic should not be able to provide a correct value-estimation for the stochastic policies when calculating the policy gradient. The paper should give a further explanation and theoretical analysis of this approach. Secondly, in Algorithm 1, the centralized critic is first updated k iterations using the “same” sampled data. Then, the question here is why not just tune the learning rate? Theoretically, this k iterative updates seems ok, but if the goal is to obtain a critic for providing more precise estimation, then why not just keep training until it converges? Lastly, if you have the k iterations for LICA, do you also implement the same setting for other policy gradient baselines such as COMA, MADDPG, SQDDPG, IDDPG, and LIAR? If not, then the comparison shown in the results is not fair.   Also, in order to prove the advantage of having the mixing network rather than a MLP, the network architecture should be modified and everything else should be kept the same. According to the results shown in Fig. 2b and Fig. 2c, the values of lambda (0.03, 0.04) with MLP are also different from the values (0.06, 0.09) used with the mixing network. This choice should be justified. Regarding the StarCraft results, the average performance of each method is only over three independent runs which is not enough. If you look into the original COMA, QMIX, and MAVEN papers, they conducted 35 runs, 20 runs, and 12 runs respectively.  Moreover, an essential phenomenon being shown in Fig 4e is also showing that it is necessary to perform more runs to make the results more convincing, because the learning curves of COMA, QMIX and VDN have huge differences with the results shown in the paper “The StarCraft Multi-Agent Challenge”. In addition, MAVEN could be a reasonable baseline to compare with, since it also focuses on improving exploration. 

Correctness: The paper appears correct, but some issues should be clarified. Some are mentioned above and below. Additionally, it isn't clear why the proposed objective maximizes the returns from every state. So, the subscript t of state s in Eq. 2 should be clarified.

Clarity: The paper is generally clear, but some more formal details are needed. For instance, it is not very clear how the policy gradient is exactly computed for each policy. There should be an equation detailing this.  Also, it is not clear what the maximum time-step is for the cooperative navigation domain, which makes the results shown in Fig 3b hard to interpret. If the default horizon 25 time-step is being used as the only terminal condition, -5 episodic return (the mean performance) means the average distance between each landmark and its nearest agent through the entire episode is 5/25/3=0.06. This means each agent’s location is initialized very close to a corresponding landmark. These details should be clarified.

Relation to Prior Work: The architecture of LICA borrows some the ideas from MADDPG and QMIX and this could be discussed in more detail.

Reproducibility: No

Additional Feedback: ******* After discussion and author feedback ******* The author feedback was appreciated as they clarified some of the issues. More detailed responses are below. Q1. A clear definition of the Q-value function being learned is very essential in a proposed RL algorithm. Training an action-probability (AP) conditioned Q-value function using the data only in specially cases of AP with probability 1 means the Q-value function is only able to provide a good estimation for deterministic policies, while the proposed algorithm uses the Q-value function to calculate the objective for optimizing the policies, conditioning on the APs output by stochastic polices, which is problematic. Also, it is not clear which exactly literatures the author mentioned above, and it would be nicer to include the corresponding titles. Q2. This argument is not convincing. Please clarify in the paper. Q3. OK, but it seems unlikely that k=1 would work best for the others. Please clarify. Q4. Line 296 in the paper doesn’t provide a good clarification. Keeping lambda the same and only modifying the network architecture is a more solid comparison. Regardless, please clarify in the paper. Q5. Please refer to the number of runs in other papers I mentioned in my review. More runs are certainly needed to make the results more convincing. Q6. This is good. Q7. Please formally justify your implementation choice.


Review 2

Summary and Contributions: The article presents a new MERL algorithm for cooperative joint action in the family of centralised training and decentralised execution. The algorithm produces a Q-network for the joint action that mixes the environment state and the individual policy logits in a better way from previous algorithms. In addition, they introduce a novel entropy cost regularisation that they argue improves exploration throughout training.

Strengths: The article is very well presented and contextualised. The experiments are compelling, and the narrative flow is of high caliber. The interpretation of the algorithm as credit assignment is quite interesting, and, while I'm not 100% convinced that this is exactly what is going on with this algorithm, I think it is arguable that the authors have a valid formulation.

Weaknesses: At its heart, this article is a slightly modified entropy cost term, and a slightly transformed joint-action critic. The authors talk about possible ready extensions to continuous domains, but no concrete evidence is provided.

Correctness: The methodology is of very high quality. The experiments are well motivated, and sufficient to illustrate the strengths of the proposed algorithm. The level of difficulty chosen is commendable, and the comparison against many other SoTA algorithms is very good to see.

Clarity: This is definitely one of the strongest points of the article, where the contextualised choices for the new components are presented clearly and at the correct level of detail. The results are discussed adequately in the text, and the intuitive interpretations accompanying them are topical, without edging on wild speculation.

Relation to Prior Work: The work is very well contextualised.

Reproducibility: Yes

Additional Feedback: I particularly appreciate the use of open source implementations throughout. Thanks!


Review 3

Summary and Contributions: The authors designed a new critic structure for implicit multi-agent credit assignment. Compared with the vanilla critic of MADDPG, the Mixing Critic in LICA decouples the gradients of actions and state and provides more state information to the policy gradients. To keep consistent exploration, they also proposed adaptive entropy regularization, which dynamically rescales the entropy by dividing a measure of policy stochasticity. The authors compared LICA against other baselines in MPE, SMAC, and Traffic Junction, and verified the effects of the proposed components by ablation studies.

Strengths: The paper is generally clear and well-structured. I much agree with the motivation that credit assignment may not require an explicit formulation. Explicit credit assignment is hard to compute and would be unrealistic in complex scenarios. The adaptive entropy regularization is well-motivated, simple, and practical, which allows easier tuning and balances exploration and exploitation during training. The empirical results on various tasks and comparisons to baselines are well done. The visualizations of the entropy term show that adaptive entropy regularization encourages consistent exploration.

Weaknesses: My main concerns are the novelty and benefits of the Mixing Critic. I think the main difference between LICA and the single-critic variant of MADDPG is that LICA formulates the critic as a hypernetwork. But what you analyzed in Discussion (Page 4), that the policy gradients are decoupled from the state update and carry the state information, do not necessarily lead to better credit assignment. The policy $\theta$ is updated by $\frac{\partial Q(s,a)}{\partial a}\cdot \frac{\partial a}{\partial \theta}$, where $\frac{\partial a}{\partial \theta}$ is unrelated to the critic. The rightness of $\frac{\partial Q(s,a)}{\partial a}$ is determined by how accurate the learned function $Q(s,a)$ is. If the MADDPG critic learns a good approximation of $Q(s,a)$, it could also offer the right direction of action vector, whatever the direction is decoupled from the state update or not. We only need the gradients of action on the given state without actually updating the state. Moreover, the gradients provided by MADDPG critic also contain the state information, since the state is necessary to compute $Q(s,a)$ and gradients. For example, for $y = Activation(wx+b)$, b will influence $\frac{\partial y}{\partial x}$ by influencing the Activation (relu,tanh). (3) and (4) in the Discussion have been achieved in MADDPG, which cannot be seen as the contributions of LICA. In the ablation experiments for Mixing Critic, LICA outperforms the single-critic variant of MADDPG. However, more explanations are expected to support the conclusions that decoupled gradients and fused state information do bring a better credit assignment. Is it possible that the Mixing Critic just learns a better approximation of $Q(s,a)$? Moreover, in practice, I find that concatenating state and actions at the beginning will lead to poor performance. Concatenating the representations of state and actions after MLP might improve MADDPG critic. The adaptive entropy regularization could adjust the levels of exploration. However, since this entropy term will be large once the policies become deterministic, which would make the policies be stochastic again, how to guarantee that LICA could converge to stable policies? In the experiments for the capacity of credit assignment, I do not think COMA is a good baseline since it is an on-policy method, but LICA and MADDPG are off-policy. I suggest comparing LICA with MAAC, an off-policy method equipped with a similar credit assignment mechanism like COMA.

Correctness: Basically correct.

Clarity: Yes.

Relation to Prior Work: Yes.

Reproducibility: Yes

Additional Feedback:


Review 4

Summary and Contributions: This paper addresses the issue of credit assignment in a multi-agent reinforcement learning setting. The paper presents an implicit technique that addresses the credit assignment problem in fully cooperative settings. The basic idea (for which the paper provides some empirical evidence) is that an explicit formulation for credit assignment may not be required as long as the (centralized) critic is designed to fuse policy gradients through a clever associate with with agent policies. To prevent premature convergence a technique called adaptive entropy regularization is presented where magnitudes of the policy gradients from the entropy term are dynamically re-scaled to sustain consistent levels of exploration throughout training. Results are shown on particle and StarCraft II environments.

Strengths: A key strength of this work is the focus on an important problem: credit assignment in a multi-agent reinforcement learning setting and an implicit strategy to deal with it. If I understand correctly, the algorithm does not require any communication between agents once training is complete. This is a significant strength.

Weaknesses: The key limitations of this work are that the results, while promising, suggest a need for further empirical evaluation before they can be firm. The results in the particle environment are not much of an improvement over past work but I accept the authors' claim that this is because the environment is not sufficiently challenging. In the Starcraft environment, I would have liked to see further training iterations and more complex settings. I accept that in the results presented there is clear evidence that LICA converges faster (except in one setting) but I am not convinced that this will carry over to more complex settings where success is an uneven mix of 'individual performance' and 'cooperation'.

Correctness: The claims and method appear to be correct and the empirical methodology is a good first step in the right directlon. At the risk of being repetitive, I will repeat what I think is the key limitation of this work - The idea behind the paper is intriguing, but the results are not convincing. Further experiments in more challenging settings are likely needed to show that this work has impact. Right now, it reads more like a report where the initial results are promising.

Clarity: The paper is well written and easy to follow.

Relation to Prior Work: The paper is well-situated in the literature, and the relation to prior work is clear. The authors also make it clear how their work differs from previous contributions.

Reproducibility: Yes

Additional Feedback: Thanks for the additional experiments on more complex environments. The results are a bit hard to make out (the figures in the rebuttal are tiny) but just about legible under magnification. Also, the comment about results on MMM2 are appreciated.

[Author Response · NeurIPS 2020]

**We appreciate all reviewers for their feedback!** We're glad that they find our methods well presented (**R2,3,4**), moti-
vated (**R3,4**), and contextualized (**R2,4**), novel (**R1,2,4**), simple and practical (**R3**), and experiments well-designed (**R2**).
**Reviewer #1** Q1. Definition of $Q$. The critic aims to estimate the joint action-value based on the **action probabilities** (AP).
As discussed in L121-128, our intuition is to train policies directly towards optimal cooperation with **full differentiability**,
and we use sampled actions (**special cases of AP with probability 1**) for critic training because the target values (defined
by action-specific rewards $r(s_t, u_t)$) over arbitrary AP are hard to estimate. In fact, similar ideas were explored for single RL
settings [Wierstra,Schmidhuber,ECML'07; Weber et al, AISTATS'19] with proper justification. We'll add more discussion
and citations accordingly. Q2. $k$ iterations critic update. Yes, $k$ is intended to give better critic estimation and tuning LR is
equivalent; it was included as a **practical generalization**: training till convergence could take long and risk overfitting, and
tuning $k$ instead of LR may avoid overshooting. Q3. $k$ for other PG baselines? Yes, e.g. $k=2$ for LICA/MADDPG and $k=1$ for
others work best empirically in SC II. We'll revise to avoid confusion. Q4. Different $\lambda$ for MLP vs mixing critic. We observed
that the **architecture change alone resulted in more stochastic joint actions**, and as clarified in L296, the choices of $\lambda$
for MLP critic ensure a fair comparison of **policy stochasticity** (Fig.2(b)) against the best LICA run ($\lambda=0.09$). We found
that setting $\lambda=0.09$ for MLP critic clearly results in over-regularization and gives even worse performance. Q5. Need more
runs/inconsistency with SMAC paper. We want to point out that our results on all maps except 2c_vs_64zg are consistent
with previous work (e.g. [3,20,21]); for 2c_vs_64zg specifically, our investigation suggests that the inconsistency is due
to a **mismatch in SC II gameplay version**: we base our experiments on the latest SMAC repo which uses **v4.10**, while
SMAC paper seems to use **v4.6** (commit history); critically, **v4.7** added changes that made Colossi units more powerful,
changing the dynamics of 2c_vs_64zg. Nevertheless, we'll add more runs for SCII as suggested. Q6. Compare with
MAVEN. As suggested, we added comparisons on 2 **Super Hard** maps in Fig.A/B. With same #iterations, **LICA performs
considerably better**. Q7. Why $t$ in $s_t$ for Eq.2? Optimizing expected returns over different $t$ is rather standard and often
implied under various notational choices; e.g. see [4,3,28] and their implementation. Q8. Eqn for per-agent policy gradients.
Due to full differentiability (L145), the PG for agent $a \propto \sum_t \nabla_{\theta_a} p_t^a \nabla_{p_t^a} Q^\pi \left( s_t, p_t^1, ..., p_t^a, ..., p_t^n \right)$ with $p_t^a = \pi_{\theta_a}^a (\cdot | z_t^a)$;
we'll update accordingly. Q9. Details of MPE. For Fig.3(b,c), we use 200 steps (L214), -1 reward for every pairwise
collision, and we report the **mean reward over all timesteps and agents** in each episode. We'll clarify the metrics in the
paper; see also our base repos [13,28]. Q10. Add discussions for QMIX/MADDPG. Thanks! We'll update accordingly.
**Reviewer #2** *Thanks for recognizing our work!* Q1. LICA in continuous domains. While this is a future extension, we
emphasize that LICA doesn't pose extra constraints on top of previous work [4,9,13] that readily handles continuous actions.
**Reviewer #3** Q1. Benefits/novelty of mixing critic. Let us consider the generalization where both MLP critic ($C_{MLP}$) and
mixing critic ($C_{Mix}$) operate on representations of states and actions $f_s(s)$, $f_a(a)$. Then, in both cases, we have $\frac{\partial Q}{\partial a} = \frac{\partial Q}{\partial h} \frac{\partial h}{\partial a}$,
where $h = f_s(s) + f_a(a)$ for $C_{MLP}$ and $h = f_s(s) f_a(a)$ for $C_{Mix}$ is the **first mixed representation** of $s,a$ before activation (i.e.
after concat+linear for $C_{MLP}$ and before $\sigma(\cdot)$ for $C_{Mix}$, Fig.1(b)). Since $g(h)=Q$ is non-linear/non-interpretable in both cases,
the crucial difference is thus that $\frac{\partial Q}{\partial a} = \frac{\partial Q}{\partial h} \frac{\partial h}{\partial f_a} \frac{\partial f_a}{\partial a} = \frac{\partial Q}{\partial h} \frac{\partial f_a}{\partial a}$ for $C_{MLP}$ and $\frac{\partial Q}{\partial a} = \frac{\partial Q}{\partial h} \frac{\partial h}{\partial f_a} \frac{\partial f_a}{\partial a} = \frac{\partial Q}{\partial h} f_s(s) \frac{\partial f_a}{\partial a}$ for $C_{Mix}$, i.e. $C_{Mix}$
**adds an extra, direct state representation**. ...do not necessarily lead to better credit assignment (CA): While better CA is
not *guaranteed*, we argue **better utilization** of state provides a basis for better CA. Rightness of $\frac{\partial Q}{\partial a}$ ...determined by accuracy
of $Q(s,a)$...$C_{Mix}$ just learns a better $Q(s,a)$? we argue that the **composition** of $\frac{\partial Q}{\partial a}$ in $C_{Mix}$ is the key factor, and a better $Q(s,a)$,
if any, would rather be a result of it. $C_{MLP}$ also contains state...: We intend to convey that $C_{Mix}$ has a better utilization of $s$ and
will revise all inaccuracies in Sec 3.2. Discussion (3,4)...aren't contributions: We'll revise accordingly; note that they remain
valid and were discussed as LICA's *properties* rather than novelties. Concat after MLP for $C_{MLP}$: As suggested, we ran a com-
parison in Fig.C where MLPs are added before concat; results confirm our earlier analysis which covers this case. Q2. Could
LICA converge to stable policies? While we cannot provide a full analysis here, we emphasize that our empirical evidence
across different $\lambda$'s, scenarios, complexity (Fig.4(a-f)), and environments with repeated runs (Fig.3/4) suggests that policies
eventually reach a stochasticity equilibrium (Fig.2(b,c)); this may in fact sustain smoother object landscapes and aid policy
convergence [1]. Q3. Compare with MAAC. By design, the simplicity of the quoted **1-step** game obviates most key aspects
that differentiate on/off-policy learning (future estimation, separate target/behavior nets, replay buffers) and focuses only on
the **mechanism for credit assignment**. However, we appreciate your suggestion and will add this discussion accordingly.
**Reviewer #4** Q1. Improvements in MPE. We stress that compared to the

previous SOTA [28], our method achieved similar gains despite approach-
ing the limits of the selected envs. Q2. Complex settings w/ uneven mix of
'individual performance' and 'cooperation'. In fact, MMM2 (**Super Hard**,
Fig.4(f), Supp L20-23, and demo) is *precisely* one such setting where our method has **sizable advantage over others**.
Winning heavily relies on the performance of the 1 healer unit and cooperation of the 9 attack units. Q3. SC II: further
training/more complex settings. We emphasize that many previous work mainly focuses on **Easy** maps (e.g. [3,4,20]) and
lacks diversity in map choices (e.g. [3,4,20,14,ROMA ICML'20]); on our diverse maps (L252-254), **we achieved similar
or significantly more gains** compared to previous work **with similar #iterations**. At **R4**'s request, we also added results
on 2 extra **Super Hard** maps (6h_vs_8z,3s5z_vs_3s6z) in Fig. A/B, showing **sizable gains over previous methods**.
Q4. It reads more like a report. We respectfully disagree. On top of **R2**'s recognition and our above response, we'd also
highlight our comparison against SOTA in 2019 [3,25] and our extensive component studies (Sec 4.3, Supp A2, Fig.2) that
are equally or more comprehensive compared to previous work (e.g. [3,4,20,28,14]).

[Meta-Review · NeurIPS 2020]

Reviewers agree that this is a borderline paper, but overall are happy with the rebuttal and have adjusted scores slightly. There is also agreement that the paper is well-written and clear, with supported contribution, but with somehow minor algorithmic improvements. Reviewers seem ok to accept if the authors provide additional clarification in their crc as provided in the rebuttal. As an AC I am in favor of acceptance.